# Opportunities and Challenges of Small Molecule Inhibitors in Glioblastoma Treatment: Lessons Learned from Clinical Trials

**DOI:** 10.3390/cancers16173021

**Published:** 2024-08-29

**Authors:** Linde Hoosemans, Marc Vooijs, Ann Hoeben

**Affiliations:** 1Department of Radiation Oncology (MAASTRO), GROW School for Oncology and Reproduction, Maastricht University Medical Center+, 6229 HX Maastricht, The Netherlands; l.hoosemans@maastrichtuniversity.nl (L.H.); marc.vooijs@maastrichtuniversity.nl (M.V.); 2Department of Medical Oncology, GROW School for Oncology and Reproduction, Maastricht University Medical Center+, 6229 HX Maastricht, The Netherlands

**Keywords:** glioblastoma, clinical trials, small molecule inhibitors, tyrosine kinase inhibitors, resistance

## Abstract

**Simple Summary:**

Glioblastoma is the most common brain tumour, with a poor prognosis of about 15 months despite intensive treatment. Many trials have tested new drugs targeting specific genes, but none have succeeded, and treatments have not changed since 2005. This review explores why clinical trials with these drugs failed. It highlights the potential of combining different drugs to overcome resistance and suggests ways to improve future trials. The goal is to understand treatment failures and find new drug combinations to improve survival for GBM patients.

**Abstract:**

Glioblastoma (GBM) is the most prevalent central nervous system tumour (CNS). Patients with GBM have a dismal prognosis of 15 months, despite an intensive treatment schedule consisting of surgery, chemoradiation and concurrent chemotherapy. In the last decades, many trials have been performed investigating small molecule inhibitors, which target specific genes involved in tumorigenesis. So far, these trials have been unsuccessful, and standard of care for GBM patients has remained the same since 2005. This review gives an overview of trials investigating small molecule inhibitors on their own, combined with chemotherapy or other small molecule inhibitors. We discuss possible resistance mechanisms in GBM, focussing on intra- and intertumoral heterogeneity, bypass mechanisms and the influence of the tumour microenvironment. Moreover, we emphasise how combining inhibitors can help overcome these resistance mechanisms. We also address strategies for improving trial outcomes through modifications to their design. In summary, this review aims to elucidate different resistance mechanisms against small molecule inhibitors, highlighting their significance in the search for novel therapeutic combinations to improve the overall survival of GBM patients.

## 1. Introduction

Glioblastoma (GBM) is the most common malignant primary brain tumour. The 2016 CNS WHO classification defines two groups: (1) isocitrate dehydrogenase (IDH)-wildtype (wt) glioblastoma and (2) IDH-mutant (mt) grade IV astrocytoma [1]. IDH-wt GBM is mainly diagnosed in adults aged 50 to 60, and patients have a median survival of 15 months [2,3], whereas IDH-mutant grade IV astrocytoma is mostly diagnosed between the ages of 35 and 45 and is the result of dedifferentiation from a low-grade astrocytoma [2,4]. Patients with IDH-mt grade IV astrocytoma have a median survival of approximately 31 months [5]. This review will focus on IDH-wt GBM.

After diagnosis, treatment consists of maximal surgical resection, followed by radiation and chemotherapy (temozolomide) [3,6]. Even with this intensive multimodality treatment schedule, patients with GBM cannot be cured, and recurrence is inevitable for all patients [3,5]. Despite multiple trials, no innovative life-prolonging treatments have been identified for patients with IDH-wt GBM to improve the current standard of care.

One of the main reasons for the failure of the clinical trials in GBM is inter- and intra-tumoral heterogeneity on the phenotypic, genotypic and transcriptional levels. The intertumoral heterogeneity of GBM tumours is marked by four subtypes Verhaak et al. identified by genomic analysis [7]. First, the classical subtype is primarily characterised by alterations in epidermal growth factor receptor (*EGFR*). This subtype also shows a homozygous deletion spanning the *Ink4a/ARF* locus. Second, the pro-neural subtype is primarily characterised by amplification or activating the mutation of platelet-derived growth factor receptor A (*PDGFRA*) and point mutations in *IDH1* and *TP53*. Third, the neural subtype is characterised by the expression of neuronal markers such as neurofilament light chain (*NEFL*), gamma-aminobutyric acid type A receptor subunit alpha1 (*GABRA1*), synaptotagmin 1 (*SYT1*) and potassium chloride transporter member 5 (*SLC12A5*). Finally, the mesenchymal subtype is primarily characterised by the loss of neurofibromatosis type I (*NF1*) [7]. 

In addition to this intertumoral heterogeneity, Patel et al. found that one genotypic subtype does not characterise a GBM tumour, but rather, several subtypes occur in one tumour [8]. Additionally, intra-tumoral heterogeneity is further supported by identifying four main cellular states in which tumour cells in GBM exist: neural progenitor-like (NP-like), oligodendrocyte-progenitor-like (OPC-like), astrocyte-like (AC-like) and mesenchymal-like (MES-like) states [9]. These states are influenced by the tumour microenvironment, which has been shown to have a spatially heterogeneous immune compartment based upon the immune cell subtype and activation levels [9]. Alterations in cyclin-dependent kinase 4 (*CDK4*), *EGFR*, *PDGFRA* and *NF1* are associated with the relative frequency of cells in each state, which varies between GBM samples [9]. This inter- and intra-tumoral heterogeneity in GBM may be the reason why “one shoe fits all” conventional treatment options do not achieve long-term responses in GBM [8,9,10].

Achieving a uniform drug distribution in GBM is complex due to the intact blood–brain barrier (BBB) in some areas. The BBB, primarily composed of endothelial cells forming tight junctions, restricts drug transport from the capillaries to the brain [11,12]. Only drugs with a molecular mass under 500 Da and high lipid solubility can pass the BBB [13]. Consequently, all large molecule agents, such as gene therapeutics and monoclonal antibodies, and over 98% of small molecule inhibitors cannot cross the BBB [13]. Efflux transporters further complicate drug delivery by expelling chemotherapy and small molecule inhibitors at the BBB [14,15].

Small molecule inhibitors might offer a potential solution to the inter- and intra-tumoral heterogeneity in GBM by targeting the drivers of tumorigenesis [16]. They can bind to various intracellular and extracellular targets [17,18]. Small molecule inhibitors can either act as enzyme inhibitors, reducing enzyme activity, or as receptor antagonists, countering receptor effects [17,18]. These compounds can function as multi-kinase inhibitors, targeting a wide range of the human kinome, or as selective inhibitors, targeting a specific component of a signalling pathway [19]. By targeting different signalling pathways and cellular processes involved in cancer, small molecule inhibitors are therefore an interesting treatment option for various types of cancer [19]. In *BRAF*-mutated metastatic melanoma and *EGFR*-mutated advanced non-small cell lung cancer (NSCLC), small molecule inhibitors targeting *BRAF* or *EGFR*, respectively, significantly increased the overall survival and have become the standard of care in these patients [20,21,22].

Nonetheless, despite different trials with small molecule inhibitors in patients with GBM, none has led to an improvement in the standard of care for these patients.

Therefore, we will review trials involving small molecule inhibitors in GBM patients and discuss possible resistance mechanisms, contributing to their lack of efficacy. Furthermore, we will propose potential combinatorial therapies based on the molecular characteristics of individual tumours.

## 2. Clinical Trials with Small Molecule Inhibitors in GBM 

### 2.1. Mono-Target Small Molecule Inhibitors

Clinical trials in GBM with small molecule inhibitors specifically inhibiting one specific target are listed in Table 1.

Only a few mono-target small molecule inhibitors evaluated in phase I GBM trials have progressed to phase II trials, mostly with disappointing results (Table 1). However, most of these trials are biomarker naïve and unable to identify “on-target” effects of the specific drug, monitor treatment response or identify resistance mechanisms. This challenge arises since re-sampling of the tumour during treatment is not feasible due to its location, which could damage eloquent brain areas and result in a subsequent loss of function. Furthermore, the potential of liquid biopsy in GBM remains unexplored, as it faces several challenges, such as the extremely low concentrations of biomarkers in the blood and their short half-life, both of which complicate detection [23]. So far, no prognostic or predictive biomarkers have been identified in the blood of GBM patients [23,24]. Further studies are necessary to assess the sensitivity and specificity of liquid biopsies in GBM.

The mono-target compounds that have shown some clinical benefit, defined as an improvement in progression-free survival (PFS) and/or overall survival (OS) compared to the standard of care, in (a subset of) GBM patients included in these trials, will be discussed in more detail.

One of the most frequently altered genes in GBM is the epidermal growth factor receptor (*EGFR*). EGFR is a tyrosine kinase receptor amplified and constitutively activated in 57% of all GBM patients [4]. In fifty percent of the tumours with *EGFR* amplification, activating *EGFR* gene rearrangements occur, with the most common extracellular domain mutation being *EGFRvIII* [25,26]. This mutation leads to a deletion of exons 2–7 of the *EGFR* gene and renders the mutant receptor incapable of binding any known ligand. Despite this, *EGFRvIII* displays low-level ligand independent constitutive signalling that is amplified by reduced internalisation and downregulation.

The small molecule inhibitors erlotinib and gefitinib, both binding to the kinase domain of EGFR and inhibiting phosphorylation and activation, have been assessed in multiple trials (Table 1). Overall, the response rates were low, and overall survival did not improve significantly. Moreover, *EGFRvIII*/*EGFR* amplification did not correlate with a better outcome compared to *EGFR* wildtype [27,28,29,30].

However, in a small subset of patients, a stabilisation or even decrease in tumour volume was identified when treated with erlotinib. Haas-Kogan et al. showed, based upon a tissue analysis, that erlotinib has a significantly better effect on tumours expressing high levels of EGFR and low levels of phosphorylated protein kinase B (AKT) compared to tumours with low EGFR levels and high levels of phosphorylated AKT [31,32]. This indicates that activation might be a resistance mechanism of GBM to EGFR inhibitors. This mechanism was also suggested in a trial investigating gefitinib, which showed a significantly better PFS and OS in patients with de novo GBM with *EGFR* gene alterations combined with wildtype phosphatase and tensin homolog (*PTEN)* (inactivation of the AKT pathway) compared to patients with wildtype *EGFR* and altered *PTEN* (activation of the AKT pathway) [33]. Similarly, in *EGFR*-mutated non-small cell lung cancer (NSCLC), *PTEN* loss is known as a resistance mechanism to EGFR inhibitors [34,35].

A phase II trial also found that erlotinib treatment in patients with recurrent GBM resulted in improved six-month progression-free survival (PFS-6) and comparable median OS compared to historical values for patients undergoing treatment with irinotecan [36]. Nonetheless, because of the small number of responses, no conclusions could be drawn from the molecular subgroup analyses. Based on the limited efficacy of erlotinib and gefitinib in the total, unstratified GBM study population, these compounds are currently not approved for the standard of care in GBM patients.

In addition to the trials mentioned above, other trials have been conducted investigating mono-target small molecule inhibitors but were not able to show an effect on tumour size and therefore did not improve overall survival for GBM patients. These trials investigated the effects of buparlisib (phosphoinositide 3-kinase (PI3K) inhibitor), capmatinib (mesenchymal epithelial transition (c-MET) inhibitor), periforsine (AKT inhibitor), deforolimus (mechanistic target of rapamycin (mTOR) inhibitor), PF-06840003 (indoleamine 2,3-dioxygenase-1 (IDO-1) inhibitor), GSK2256098 (focal adhesion kinase (FAK-kinase) inhibitor), adavosertib (Wee-1 inhibitor), pegdinetanib (VEGFR-2 inhibitor), navtemadlin (mouse double minute 2 homolog (MDM2) inhibitor), tipifarnib (FTase subunit ß inhibitor), AXL1717 (insulin-like growth factor 1 receptor (IGF-1R) inhibitor) and selinexor (exportin 1 (XPO-1) inhibitor). More details on these trials can be found in Table 1.

In summary, mono-target small molecule inhibitors showed poor efficacy in GBM, with only a few trials demonstrating modest efficacy in a specific subgroup. An explanation for the ineffectiveness of mono-target small molecule inhibitors is the clonal selection of resistant tumour cells due to intra-tumour heterogeneity [8,9]. Investigation of primary GBM tumour samples has revealed that multiple receptor tyrosine kinases (RTKs) are activated within a single tumour sample [8,10,37]. Moreover, malignant cells in a GBM tumour exhibit plasticity, and a single cell is able to generate all four states [9]. This plasticity of the cells leads to changes in distribution of these states after targeting genetic drivers [9]. By inhibiting only one genetic driver—for example, *EGFR*—only cells harbouring alterations in *EGFR* will be targeted, while the other tumour cells remain unaffected, leading to sub-clonal selection and, eventually, to recurrence. 

Additionally, most trials include unstratified GBM patient populations. Consequently, by not taking biomarkers or tumour heterogeneity into account, GBM tumours are included that are already intrinsically resistant to small molecule inhibitors because of their molecular characteristics. 

Therefore, multitarget small molecule inhibitors might be more effective in tackling the inter- and intra-tumoral heterogeneity than mono-target small molecule inhibitors and prevent sub-clonal selection [9].

### 2.2. Multitarget Small Molecule Inhibitors

Clinical trials with small molecule inhibitors targeting multiple targets are presented in Table 2.

Remarkably, most trials that showed a promising result after treatment with a multitarget small molecule inhibitor involved small molecule inhibitors targeting vascular endothelial growth factor (VEGF/VEGFR). Since GBM exhibit extensive vascularity, VEGF/VEGFR is an important target [38].

Different small molecule inhibitors have been designed to target VEGF/VEGFR. For example, in a randomised phase II trial comparing axitinib, which, apart from VEGFR, also targets PDGFRß, and c-Kit, with physicians’ best choice of therapy in patients with recurrent GBM, axitinib monotherapy was found to increase the 6-month PFS and overall response rate but not OS [39].

Another trial compared treatment with axitinib alone to axitinib combined with lomustine in patients with recurrent GBM. Axitinib improved the response rate and progression-free survival in this population compared to historical controls, but the overall survival of these patients treated with axitinib alone did not improve [40].

A phase II trial investigating the effects of cediranib, a pan-VEGFR inhibitor also targeting FLT1/4, c-Kit and PDGFRβ, showed promising results in patients with recurrent GBM [41]. Since its results were comparable to historical controls, the inhibitor proceeded to a phase III trial. However, the primary endpoint of the trial (PFS prolongation) when comparing cediranib as a monotherapy and in combination with lomustine was not met. Subgroup analysis investigating the effect of the *VEGFR* baseline levels showed no significant effect after cediranib monotherapy compared to lomustine monotherapy [42]. Hence, cediranib has not been approved for GBM.

Regorafenib, which inhibits VEGFR, PDGFR, c-Kit, c-RET, Raf-1, FGFR and Abl, showed in a randomised, controlled, phase II trial a significant improvement in overall survival in patients with recurrent GBM compared to lomustine [43]. Regorafenib therefore continues to be investigated in a phase III trial, the results of which are awaited (NCT03970447). One explanation for regorafenib’s potential as an inhibitor lies in its role as a multi-kinase inhibitor, targeting not only VEGFR but also targeting kinases involved in oncogenesis and the tumour microenvironment [44].

Although fusion genes were discovered a long time ago, their significance and role in oncogenesis was not recognised until recently [45,46]. Currently, different tyrosine kinase inhibitors (TKI) are investigated in trials to target fusion genes in recurrent GBM patients [47]. In the context of GBM, fusion genes are present in approximately 30–50% of all GBM patients, and druggable fusion genes are found in approximately 4% of these patients [48,49]. Multiple fusion genes have been identified, with the most prevalent involving *MET*, *EGFR*, *FGFR*, *NTRK*, *RET* and *ROS* [47,50]. Currently, several TKIs are investigated in trials to target these fusion genes in recurrent GBM [47]. Infigratinib (FGFR 1/2/3 inhibitor) showed limited efficacy in the entire recurrent glioma patients with the *FGFR* alterations population but demonstrated a durable response in patients with *FGFR1* or *FGFR3* point mutations and those with *FGFR3-TACC3* fusions [51]. More research is necessary to clarify the efficacy of TKIs targeting fusion genes in a biomarker-driven GBM patient cohort.

In addition to the trials mentioned above, many trials have been conducted investigating multitarget small molecule inhibitors, but none were able to improve OS (Table 2). 

While there are no phase III trials with mono-target small molecule inhibitors, there have been two phase III trials with multitarget small molecule inhibitors. However, none have shown an improvement in overall survival in a phase III trial so far. 

### 2.3. Small Molecule Inhibitors Combined with Chemotherapy/Bevacizumab

An overview of all the trials involving small molecule inhibitors combined with chemotherapy or bevacizumab can be found in Table 3.

Of all these trials, only two showed an increase in PFS and OS [52,53]. In the first trial, recurrent GBM patients with VEGF overexpression and EGFRvIII mutation showed an increase in the response rate and PFS after treatment with erlotinib combined with bevacizumab. However, since the trial only included four patients, no conclusions could be drawn [52]. In addition, other trials involving erlotinib combined with bevacizumab have also not shown an increase in PFS or OS [54,55]. However, these other trials were not biomarker-driven.

In the second trial, anlotinib, a multitarget small molecule inhibitor with similar targets as regorafenib, showed promising results in a phase II single-arm trial in twenty-one recurrent GBM patients in combinations with temozolomide with an increase in the response rate, PFS (7.3 months (95% CI 4.9–9.7)) and median OS (16.9 months (95% CI 7.8–26.0)) compared to historical controls [53].

Additionally, different trials have investigated the effects of different small molecule inhibitors combined with chemotherapy or bevacizumab. Lomustine has been investigated together with buparlisib (PI3K inhibitor), but this combination did not improve OS in 18 recurrent patients [56]. The combination of hydroxyurea with imatinib and vatalanib (VEGFR/FLT inhibitors) also did not show a positive effect on the OS of these patients [57,58]. Trials investigating the combination of irinotecan with CT-322 (VEGFR-2 inhibitor) or sunitinib (VEGFR/PDGFR inhibitor) were unsuccessful as well [59,60]. Other small molecule inhibitors have been investigated in combination with temozolomide or bevacizumab, but almost all of these trials could not show an increase in OS (Table 3). 

### 2.4. Combinations of Small Molecule Inhibitors

A method to target intra-tumoral heterogeneity is to target multiple kinases in the same tumour. Only a limited number of trials have investigated the combinatorial treatments of different small molecule inhibitors. These trials are listed in Table 4. 

A trial combining gefitinib and cediranib, targeting EGFR and VEGFR, showed a trend towards an improved response rate in recurrent GBM patients in a phase II trial [61]. Unfortunately, this trial was discontinued prematurely, and no statistically significant analysis of the PFS and OS was performed. 

The *BRAF V600E* mutation is found in approximately 3% of GBM patients [62]. A single-arm, phase II basket trial that was focused on *BRAF V600E*-mutated glioma showed an increase in the response rate ((32% (95% CI 17–51)) and OS (13.7 months (95% CI 8.4–25.6)) in 31 patients with recurrent *BRAF V600E*-mutated GBM after they were treated with dabrafenib, a BRAF V600E inhibitor, combined with trametinib, a MEK inhibitor [63]. This combination therefore makes an interesting treatment option for this specific GBM subgroup. 

Most of the trials combined different small molecule inhibitors with a mTOR inhibitor. The PI3K/Akt/mTOR pathway is frequently altered in GBM, making it an interesting target [64]. As discussed above, targeting this pathway is not effective as a monotherapy (Table 1 and Table 2). Combinations with erlotinib or gefitinib and sirolimus or temsirolimus (mTOR inhibitor) and temsirolimus with perifosine (AKT inhibitor) or sorafenib (Raf/VEGFR/PDGFR inhibitor) have been investigated. None of these trials, however, showed an increase in the PFS, OS or response rate (Table 4). This is most likely due to the fact that the majority of PI3K/mTOR inhibitors are unable to pass the BBB and achieve adequate concentrations at the tumour site [65,66].

In addition, researchers have investigated combinations of different (multitarget) small molecule inhibitors, targeting various pathways, none of which showed an increase in OS in GBM patients (Table 4).

## 3. Discussion

### 3.1. Intrinsic Versus Acquired Resistance

The discouraging results of trials with small molecule inhibitors in patients with GBM underscore the complexity of developing a treatment schedule for these patients. Multi-omics analysis has unravelled the complexity of proteomic, phosphorylation, metabolic, lipidic and immunogenic alterations and their influence on GBM subtypes and survival; yet, despite this progress, a uniformly present and druggable driver oncogene across all GBM tumour remains elusive, making a “one show fits all” treatment targeting all GBM subclones unavailable for these patients [4,7,8,9,67,68].

Extensive research has been conducted investigating the effects of mono-target small molecule inhibitors (Table 1), but most of these trials did not show an improvement in PFS and/or OS. One of the possible reasons for failure of these trials is the lack of patient stratification. In two studies, subgroup analyses demonstrated that patients with amplified *EGFR* and inactivation of the AKT pathway responded better to EGFR inhibitors than those without [31,32,33]. Additionally, the investigation of primary GBM tumour samples has revealed that multiple RTKs are activated within a single tumour sample [8,10,37]. The inhibition of only one genetic driver will lead to the clonal selection of resistant tumour cells and, eventually, recurrence.

In addition to inter- and intra-tumoral heterogeneity and the clonal selection of resistant tumour cells, other mechanisms may contribute to the primary resistance to small molecule inhibitors in GBM. First, while the majority of small molecule inhibitors primarily target the kinase domain of a receptor, the situation differs in GBM, where mutations may not always be confined to this domain. An illustration of this variability is observed in the case of the EGFR receptor, which most prevalent mutation, the *EGFRvIII* mutation, is situated in the extracellular domain of the EGFR receptor, resulting in the constitutive activation of non-mutated kinase domains [25,26]. Third-generation EGFR inhibitors, like Osimertinib, used in *EGFR*-mutated non-small cell lung cancer, specifically targeting the mutated kinase domains, are ineffective in *EGFR*-amplified or *EGFRvIII*-mutated GBM [25,26].

Lastly, GBM tumours can bypass the effect of a single small molecule inhibitor by activating other pathways. These bypass signalling pathways could be another explanation for treatment resistance, which supports the need for combination therapies.

In GBM, *ERBB3*, *IGFR1R* and *TGFBR2* were positively correlated with *PDGFR*, and combining a PDGFR inhibitor with either an ERBB3 inhibitor or an IGF1R inhibitor resulted in a more significant reduction in tumour growth than each inhibitor alone [69].

Moreover, the upregulation of c-MET has been found after EGFR monotherapy in glioblastoma stem cells (GSCs) in vitro, causing long-term self-renewal ability in these cells [70]. The inhibition of both MET and EGFR resulted in overcoming the resistance to an EGFR inhibitor (gefitinib) in a preclinical mouse model [71]. Further research, however, found that, after EGFR and c-MET inhibition, ERK was reactivated via a NF-kB-dependent feedback loop that led to the activation of FGFRs. This bypass resistance mechanism could be overcome in vivo through simultaneously inhibiting FGFR [72]. More research has demonstrated that the expression of EGFRvIII induces the transactivation of c-Jun N-terminal kinase isoform 2 (JNK2) in GBM cells, which, in turn, activates the hepatocyte growth factor (HGF)/c-MET signalling pathway [73].

These preclinical results demonstrate crosstalk between different RTKs but also highlight that the inhibition of three or more RTKs is essential to overcome this resistance mechanism. While this approach is feasible in preclinical settings, using a combination of three or more mono-target small molecule inhibitors in patients could lead to severe toxicity, hindering clinical application. Consequently, targeting downstream feedback loops may circumvent the need for multiple inhibitors, thereby reducing multi-drug toxicity while maintaining a high efficacy in overcoming treatment resistance in GBM.

Crosstalk between the PI3K/mTOR pathway and MEK/ERK pathway is one of these feedback loops. Inactivation of the PI3K/mTOR pathway activates the MEK/ERK pathway and vice versa, thereby bypassing the effect of either PI3K/mTOR inhibitors or MEK/ERK inhibitors [74,75]. The combinatorial treatment of a MEK inhibitor with a dual PI3K/mTOR inhibitor was therefore found as a synergistic combination treatment in vitro [74]. However, the synergistic efficacy in vivo was minimal, most likely due to subtherapeutic dosing due to dose-limiting toxicity and poor tumour penetrance [76]. Trials exploring combinations of the MAPK pathway and PI3K/mTOR pathway have proven unsuccessful due to their low efficacy because of low tumour penetrance and high toxicity (Table 4) [77,78,79]. Therefore, novel brain-penetrant PI3K and mTOR inhibitors, such as paxalisib, are of interest and are currently being evaluated.

Even though most trials involving multitarget small molecule inhibitors have been ineffective (Table 2), a few inhibitors tend to show some promising results, all targeting VEGFR. Regorafenib and anlotinib both showed an improvement in OS, but in both trials, the treatment group exhibited more favourable prognostic factors, potentially leading to an overestimation of the observed improvement in OS [39,40,43,53].

An explanation why regorafenib and anlotinib showed better results than axitinib might be that axitinib only targets VEGFR, while regorafenib and anlotinib target VEGFR, PDGFR and FGFR [44,53]. PDGFR and FGFR are also involved in angiogenesis and upregulated upon VEGFR inhibition, thereby bypassing the inhibitory effect of the VEGFR inhibitor [80,81,82,83]. The inhibition of all three pathways is therefore more effective than inhibiting VEGFR alone [39,40,44,53].

Moreover, anlotinib was more effective than regorafenib, which could be explained by the fact that anlotinib was combined with TMZ in GBM. Preclinical research has shown that anlotinib combined with TMZ decreased GBM growth more than anlotinib alone via inhibition of the JAK2/STAT3/VEGFA pathway [84]. This illustrates how the concurrent administration of chemotherapy with, in particular, an angiogenesis inhibitor can enhance the efficacy. However, a combination of bevacizumab with lomustine in recurrent GBM failed to show a survival advantage in a phase III trial [85].

In other types of solid cancer, such as colorectal carcinoma, the inhibition of bevacizumab/VEGFR also failed to show an improvement in OS [86]. This indicates that targeting angiogenesis alone merely slows tumour growth and the tumour can circumvent the inhibition, which results in an increase in PFS but not in OS.

Different combinatorial treatments with small molecule inhibitors have been investigated, most of them with disappointing results (Table 4). One trial investigating the combination of dabrafenib with trametinib in recurrent *BRAF V600E*-mutated GBM patients showed an increase in OS [63]. Dabrafenib combined with trametinib is currently considered the standard of care in progressive BRAF V600E-mutated CNS tumours [87]. Most likely, this combination is successful, since, by adding a MEK inhibitor, the MAPK pathway is inhibited at two positions.

### 3.2. Influence of the Tumour Microenvironment

The likelihood of a combination treatment exclusively targeting tumour cells curing GBM is low, since the tumour microenvironment (TME) also plays a significant role in therapy resistance within GBM.

The TME in a GBM tumour contains not only tumour cells but also non-neoplastic cells, including immune cells, vascular cells and other glial cells [88]. GBMs are considered as immunogenic cold tumours due to low numbers of tumour-infiltrating lymphocytes (TILs) and other immune effector cell types [89].

One of the critical immunosuppressive factors in the GBM microenvironment is hypoxia, which suppresses the antitumour functions of immune cells that are able to infiltrate into the GBM microenvironment. Vice versa, functions of immunosuppressive cells, such M2 macrophages, are enhanced by hypoxia [90,91].

Trials investigating the currently available immune checkpoint inhibitors (ICIs) in newly diagnosed and recurrent GBM have not demonstrated an improvement in overall survival [92,93,94]. A key factor limiting the effectiveness of these inhibitors is the very low number of TILs in the GBM microenvironment [89,95].

Consequently, future research must focus on targeting the GBM-specific immune environment, which is characterised by immunosuppressive tumour-associated macrophages (TAM). To achieve this, it is imperative to elucidate and understand the interplay between the GBM tumour and TAMs. Additionally, more information is required on how inhibitors might be able to shift the TME to become immune-permissive. The combination of Lenvatinib with anti-PD-1 antibody and cabozantinib with a poxviral-based cancer vaccine caused a more permissive TME with a decrease in TAMs and activation of the type-I interferon response when investigated in other types of cancer [96,97].

Preclinical research in GBM has revealed that combining an EGFR inhibitor with a mTOR inhibitor can modulate the TME by downregulating immunosuppressive chemokines and inhibiting tumour-promoting macrophage infiltration [98]. Thus, small molecule inhibitors could possibly be used with a novel intent to create a shift towards an immune-permissive TME. This approach might lead towards innovative combinations of small molecule inhibitors with immunotherapy, potentially preventing the clonal expansion of resistant tumour clones. Further research is needed to explore whether such combinations of immunotherapy with small molecule inhibitors will be effective and overcome treatment resistance in GBM.

### 3.3. Clinical Trial Design

In addition to the reasons mentioned above, the disappointing results observed in trials involving small molecule inhibitors for glioblastoma patients could be attributed to several other factors related to the trial design.

The BBB makes it difficult for different drugs to arrive in effective concentrations at the tumour site in brain cancer [12,13]. The aim of phase 0 trials is to investigate the biological effect of a study drug in human patients by using a pharmacologically active but non-therapeutic dosage [99]. However, in brain tumours, a higher systemic dose will be necessary to detect study drug levels in the CNS with greater risks of related side effects [100]. Different techniques to open the BBB, such as focused ultrasound (FUS), are being investigated. These techniques might be a way to enhance drug delivery at the tumour site without the need to increase the systemic dose [101,102,103].

The advantage of the study design of phase 0 trials is that it allows for the examination of different patient samples over time, such as blood, cerebrospinal fluid and tumour tissue, hereby giving information on pharmacokinetics and -dynamics, BBB passage and target inhibition [99,100,104]. On the one hand, this information will be crucial to eliminate drugs with low tumour penetration or target inhibition and prevent unnecessary burdens on patients in phase I and II trials. On the other hand, newly developed drugs that are able to pass the BBB and inhibit the desired target effectively can be taken into phase I and II trials to investigate their effect on tumour growth, PFS and OS.

Additionally, the absence or limited use of molecular markers in patient selection might contribute to the failure of these trials, given the considerable intertumoral heterogeneity. Therefore, to find an effective drug in GBM patients, it may be necessary to select patients based on the molecular characteristics of the tumour. However, to date, only a few biomarker-driven trials have been performed using specific molecular markers of the parental tumour as an inclusion criterion. Moreover, in CNS tumours, the predictive significance of these mutations remains insufficiently studied [87].

Whole genome sequencing (WGS) is gaining prominence in oncological care. Compared to commonly used DNA diagnostic approaches, WGS offers a high sensitivity and accuracy, and all important mutation types may be consistently identified in a single assay [105]. The importance of WGS is highlighted in a study wherein 62% of the patients with different metastatic cancers had mutations that could be used for further treatment or inclusion in trials after using WGS [106].

The first biomarker-driven trial was the IMPACT study, where patients with advanced metastatic cancer that received tumour genetic profiling and were evaluated for investigational therapy were included. In total, the overall response rate (ORR) was 27% for the patients treated with matched targeted therapy and 5% for patients with non-matched therapy, emphasising the potential of biomarker-driven studies [107]. Hereafter, other biomarker-driven trials followed, such as the SHIVA trial, IMPACT-II and NCI-MATCH, to investigate the benefits of using genetic alterations to guide matched targeted therapies, aiming to enhance the ORR [108,109,110].

Given the absence of FDA approval for the use of all targeted drugs in all types of cancers, the TAPUR trial investigated the effects of FDA-approved drugs outside their approved indications in different cancer types based on pre-specified genomic targets [111]. Encouragingly, most treatment arms have shown success, underscoring the potential of this approach [111]. Similarly, the ongoing DRUP (Drug Rediscovery Protocol, NCT02925234) study, an ongoing biomarker-driven, multi-basket trial in the Netherlands, allocates patients to different treatments based on the mutational status of their tumours, achieving an overall clinical benefit rate of 33% for both rare and non-rare cancers [112].

These results highlight the importance of performing molecular tests to guarantee that all cancer patients have an equal chance of receiving the right treatment.

By identifying actionable mutations and enabling access to a broader spectrum of treatment options, studies like DRUP aim to offer renewed hope and extended survival prospects for patients with refractory GBM.

## 4. Conclusions and Future Perspectives

Although small molecule inhibitors have become part of the standard of care in various types of cancer, the overall results of clinical trials in GBM patients have been disappointing. The main factor contributing to this poor performance is the pervasive inter- and intra-tumoral heterogeneity characteristics of GBM. This tumour heterogeneity undermines the feasibility of a universal therapeutic approach, emphasising the inefficacy of a “one-size fits all” drug for these patients. Moreover, various aspects of the TME can counteract the effectiveness of small molecule inhibitors. Consequently, a need arises for further research directed towards identifying drug combinations that can target tumour clones that drive tumour relapse and are capable of modulating the complex dynamics of the TME.

In addition, more phase 0 trials will be needed to investigate drug penetration and target inhibition at the tumour site of GBM of small molecule inhibitors. Furthermore, future trials should focus on patient stratification based on molecular tumour markers to achieve the best possible results and identify drugs and drug combinations for certain patient subpopulations.

## Figures and Tables

**Table 1 cancers-16-03021-t001:** Clinical trials involving mono-target small molecule inhibitors.

Compound	Target	PMID	Trial	De Novo/Recurrent	Study Population	PFS/OS	Response Rate	Biomarker Analysis
Adavosertib (AZD1775)	Wee1	29798906	Phase 0	Recurrent	GBM *n* = 20	NA	NA	NA
Buparlisib	PI3K	30715997	Phase II	Recurrent	GBM *n* = 65	PFS 1.7m (95% CI, 1.4–1.8). OS 9.8m (95% CI, 8.4–12.1)	CR 0%, PR 0%, SD 42%, PD 54% DRC 43.8% (95% CI 31–58%)	No statistically significant association was found between *PTEN*, *PIK3CA*, *PIK3R1*, *EGFR*, *PDGFRA*, *IDH1/2* and *TP53*, and PFS6 or OS. No statistically significant association in PFS between *PIK3CA*/*PIK3R1*-mutant or *PTEN* mutant *PIK3CA*/*PIK3R1*-wildtype or *PTEN* wildtype.
Capmatinib	c-MET	31776899	Phase II	Recurrent	GBM *n* = 10, altered PTEN status	PFS not assessed due to insufficient sample size	CR 0%, PR 0%, SD 30%	NA
Deforolimus	mTOR	22037923	Phase I	Recurrent	Grade IV malignant glioma *n* = 3	NA	SD 33% as best response	NA
Erlotinib	EGFR	20150372	Phase II	Recurrent	GBM *n* = 42	PFS 2m OS 6m	CR 0%, PR 0%, SD 7.1%, PD 62%	NA
		22946346	Phase I + II	Recurrent	GBM *n* = 8	PFS 1.9m OS 6.9m	NA	NA
		24352766	Phase II	Recurrent	GBM *n* = 40, EGFR or PTEN-mutated	PFS 3.9m (95% CI 1.6–6.1) OS 7m (95% CI 1.41–4.7)	CR 0%, PR 7%, SD 21%, PD 72%	NA
		20615922	Phase II	Recurrent	GBM *n* = 48	PFS6 20% (95% CI 10.0–32.4) OS 9.7m (95% CI 5.9–11.6)	CR 2.1%, PR 6.3%, SD 33.3%, PD 54.2%	No conclusion could be draw from molecular subgroup analyses due to low response rate. EGFR amplified: OS 8.3m (95% CI 4.1–10.7) CR 4.3%, PR 4.3%, SD 43.5%, PD 47.8% Non-EGFR amplified: OS 10.6m (95% CI 4.7–14.1) CR 0%, PR 8.0%, SD 43.5%, PD 60.0%
		19204207	Phase II	Recurrent	GBM *n* = 110	PFS 1.8m OS 7.7m	NA	EGFRvIII was correlated with poor PFS in the erlotinib arm (*p* = 0.003). EGFR amplification was significant for poor outcome in the entire study population (*p* = 0.048).
Gefitinib	EGFR	29492119	Phase II	De novo	GBM *n* = 40	PFS 6m OS 14m	NA	PFS and OS were significantly (*p* = 0.005) higher in EGFR +ve/PTEN-ve compared to EGFR-ve/PTEN+ve with 9 months versus 6 months, and 20 months versus 13 months, respectively.
		14638850	Phase II	Recurrent	GBM *n* = 53	EFS 8.1w (95% CI, 7.9–9.1) OS 39.4w(95% CI, 24.3–59.4)	CR 0%, PR 0%, SD 42%, PD 58.4% within 2 months. PD 96.2% at end of follow-up	Epidermal growth factor receptor expression did not correlate with either EFS or OS.
		17353924	Phase II	Recurrent	GBM *n* = 16 AO *n* = 3 AA *n* = 9	PFS 8.4w OS 24.6w	DCR 12.5% (95% CI 1.6–38.4%). SD 12.5%	EGFR expression or gene status, and p-Akt expression predict activity of gefitinib.
		20510539	Phase II	De novo	GBM *n* = 96	PFS-1year 16.7%, OS-1year 54.2%	NA	Clinical outcome was not affected by EGFR amplification or EGFRvIII mutation.
		21471286	Phase II	Recurrent	GBM *n* = 22	OS 8.8m	NA	No difference between patients with an amplified or a normal EGFR status.
GSK2256098	FAK kinase	29788497	Phase I	Recurrent	GBM *n* = 13	PFS 5.7w (95% CI 3.1–8.3)	SD 27%, PD 73%	NA
Navtemadlin (AMG-232)	MDM2	31359240	Phase I	De novo and recurrent	GBM *n* = 10, p53wt	NA	SD 60%	NA
Pegdinetanib (CT-322)	VEGFR-2	25388940	Phase II	Recurrent	GBM *n* = 63	PFS 1.8m	1mg/kg: ORR 14.3% 2mg/kg: ORR 3.8%	NA
Perifosine	Akt	31325145	Phase II	Recurrent	GBM *n* = 16	PFS 1.58m (95% CI 1.08–1.84) OS 3.68m (95% CI 2.50–7.79)	SD 12.5%, PD 75%	NA
PF-06840003	IDO-1	32436060	Phase I	Recurrent	GBM *n* = 14 AA gr III *n* = 2 AO gr III *n* = 1	PFS 1.9–2.8m	DCR 47%	NA
Picropodophyllin (AXL1717)	IGF-1R	29113409	Phase I/II	Recurrent	GBM *n* = 8 Gliosarcoma *n* = 1	PFS 8w OS 15w	CR 0%, PR 11.1%, SD 44.4%, PD 44.4%	NA
Rapamycin	mTOR	18215105	Phase I	Recurrent	GBM *n* = 14, PTEN-deficient	No PFS or OS reported	NA	NA
Selinexor	XPO-1	34728525	Phase II	Recurrent	GBM *n* = 76	Arm B: PFS6 10% (95% CI 2.67–35.4) OS 10.5m (95% CI, 4.9–17.0) Arm C: PFS6 7.7% (95% CI 1.2–50.6) OS 8.5m (95% CI, 7.3–not evaluable) Arm D: 17.2% (95% CI, 7.78–38.3) OS 10.2m (95% CI, 7.0–15.4)	Arm B: ORR 8.3%, SD 25%, PD 62.5% Arm C: ORR 7.7%, SD 30.8%, PD 61.5% Arm D: ORR 10%, SD 23.3%, PD 56.7%	Patients with mutations in pancreatic and duodenal homeobox 1 (*PDX1*), E1A Binding Protein P400 (*EP400*) or Dedicator of Cytokinesis 8 (*DOCK8*) survived longer than patients with wildtype tumours
Tipifarnib	FTase subunit ß	16877733	Phase II	Recurrent	GBM *n* = 67	Non-EIAED: PFS 9w (95% CI 7–14) EAIED: PFS 6w (95% CI 4–8w)	CR 0%, PR 7.5% Non-EIAED: PR 11% EAIED: PR 3%	NA
Trotabresib	BET	36455228	Phase I	Recurrent	GBM *n* = 19 AA *n* = 1	PFS 1.9m (95% CI 1.4–3.4) IDH-wt: PFS 3.0m (95% CI 1.4–3.6)	SD 41%, PD 59%	NA
Vismodegib (GDC-0449)	SMO	36581779	Phase 0/II	Recurrent	GBM *n* = 41	PFS 2.3m (95% CI 1.9–2.6) OS 7.8m (95% CI 5.4–10.1)	CR 0%, PR 0%, SD 25.8%, PD 74.2%	NA

GBM: glioblastoma, AO: anaplastic oligodendroglioma, AA: anaplastic astrocytoma, MG: malignant glioma, PFS: progression-free survival, OS: overall survival, EFS: event free survival, CR: complete response, PR: partial response, SD: stable disease, PD: progressive disease, DCR: disease control rate, ORR: objective response rate. EIAED: enzyme-inducing antiepileptic drug. m: months and w: weeks.

**Table 2 cancers-16-03021-t002:** Clinical trials involving multitarget small molecule inhibitors.

Targets	Compound	Targets Specified	PMID	Trial	De Novo/ Recurrent GBM	Study Population	PFS/OS	Response Rate	Biomarker Analysis
Tumourcell	Abemaciclib	CDK 4 & 6	27217383	Phase I	Recurrent	GBM *n* = 17	NA	SD 17.6%	NA
	Afatinib	EGFR, ERBB2, ERBB4	25140039	Phase I + II	Recurrent	GBM *n* = 119	PFS 0.99m (*p* = 0.032) OS 9.8m (*p* = 0.386)	DCR 36.6% (95% CI 22.1–53.1). CR 0%, PR 2.4%, SD 34.1%, PD 34.1%	EGFR vIII+ tumours showed higher PFS versus EGFRvIII- tumours.
	Bortezomib	20S proteasome	20213332	Phase I	Recurrent	GBM *n* = 51 AA *n* = 8 AO *n* = 3 Other *n* = 4	PFS 2.1m (95% CI 1.7–2.8) OS 6.0m (95% CI 3.9–7.4)	ORR 3%, CR 0%, PR 3%, SD 23%	NA
	Cilengitide	Integrins ανβ3 and ανβ5	17470857	Phase I	Recurrent	GBM *n* = 37 AA *n* = 11 AO *n* = 1 Mixed AG *n* = 2	OS 5.6m (95% CI 4.3–8.4)	ORR 9.8%, CR 3.9%, PR 5.9%, SD 31.4%	NA
			18981465	Phase II	Recurrent	GBM *n*=81	500 mg/d: TTP 7.9w (95% CI 7.7–15.6) OS 6.5m (95% CI 5.2–9.3) 2000 mg/d: TTP 8.1w (95% CI 7.9–15.0) OS 9.9m (95% CI 6.4–15.7)	500 mg/d: ORR 5%2000 mg/d: ORR 13%	NA
			21739168	Phase II	Recurrent	GBM *n* = 26	PFS 8w (95% CI 4–16)	NA	NA
	Dacomitinib	EGFR, ERBB2 and ERBB4	28575464	Phase II	Recurrent	GBM *n* = 30, EGFR amplification GBM *n* = 19, EGFR amplification and EGFRvIII mutation	PFS 2.7m (95% CI 2.3–3.1) OS 7.4m (95% CI 5.6–9.2)	CR 2%, PR 4.1%, SD 24.5%, PD 61.2%	EGFR amplification without EGFRvIII mutation: PFS 2.7m, OS 7.8m. CR 3.3%, PR 3.3%, SD 26.7%, PD 56.7% EGFR amplification with EGFRvIII mutation: PFS 2.6m, OS 6.7m. CR 0%, PR 5.3%, SD 21.1%, PD 68.4%
	Dasatinib (BMS-354825)	Abl1, Src, c-Kit, Lck, Yes, induces autophagy	25758746	Phase II	Recurrent	GBM *n* = 50	PFS 1.7m (95% CI 1.3–1.9) OS 7.9m (95% CI 5.6–10.2)	CR 0%, PR 0%, SD 24%, PD 72%	NA
	Enzastaurin	PKCβ, PKCα, PKCγ and PKCε Chk1/Chk2	20150385	Phase I/II	Recurrent	GBM *n* = 57	PFS 1.3m OS 4.6m	ORR 30%, PR 3.5%	NA
			20124186	Phase III	Recurrent	GBM *n* = 174	PFS 1.51m OS 6.60m	ORR 2.9%, SD 38.5%, PD 41.4%	NA
	Lapatinib	EGFR, ERBB2	19499221	Phase I + II	Recurrent	GBM *n* = 17	NA	SD 23.5%	No relation between PTEN loss or EGFRvIII and outcome
	Marimastat	MMP-9, MMP-1, MMP-2, MMP-14, MMP-7	16636750	Phase II	De novo	GBM *n* = 154 Gliosarcoma *n* = 8	PFS 17.1w, OS 42.9w	NA	NA
	Palbociclib	CDK4 & 6	30151703	Phase II	Recurrent	GBM *n* = 22, Rb1-positive	PFS 5.14 weeks, OS 15.4 weeks	PD 95%	NA
	Paxalisib (GDC-0084)	PI3K, mTOR	31937616	Phase I	Recurrent	WHO grade III *n* = 14 WHO grade IV *n* = 33	NA	ORR 0%, SD 40%, PD 55%	No correlation between PTEN loss or PI3K mutations and response to GDC-0084.
			NCT03522298	Phase II	De novo	GBM *n* = 30	PFS 8.6m OS 15.9m	NA	NA
	Ribociclib	CDK4 & 6	31399936	Phase Ib	Recurrent	GBM *n* = 3, Rb+	Patient 1: PFS 2 m OS 10m Patient 2: PFS 5m OS 19m Patient 3: PFS 2m OS 12m	NA	NA
			31285369	Phase 0	Recurrent	GBM *n* = 6	PFS 9.7w OS 7.8m	NA	NA
	Romidepsin	HDAC 1, 2 (4 and 6)	21377994	Phase I/II	Recurrent	Phase I: GBM *n* = 8 Phase II: GBM *n* = 35	PFS 8w (95% CI 5–8) OS 34w (95% CI 21–47)	CR 0%, PR 0%, SD 28%, PD 72%	NA
	Sonolisib (PX-866)	PI3K (p110α), (p120γ), (p110δ)	25605819	Phase II	Recurrent	GBM *n* = 33	PFS-6 17% (95% CI 5–32%)	CR 0%, PR 3%, SD 24%, PD 73%	No statistically significant association between stable disease and PTEN, EGFRvIII, PIK3CA mutation or PIK3R1 mutation
	Tandutinib	FLT3, c-Kit, PDGFR	27663390	Phase I + II	Recurrent	Phase I: GBM *n* = 19 Phase II: GBM *n* = 30	First stage: PFS 1.9m (95% CI: 1.5–3.7) OS 8.8m (95% CI: 5.9–15.4) Time of analysis: PFS6 16%	CR 3%	NA
	Vistusertib	mTOR, PI3K isoforms α/β/γ/δ	31707687	Phase I	Recurrent	GBM *n* = 14	PFS6 26.6%	ORR 8%, PR8%, SD 38%	No correlation between pS6 status and response.
	Vorinostat	HDAC 1, 2, 3, 6, 8	19307505	Phase II	Recurrent	GBM *n* = 66	PFS 1.9m OS 5.7m	ORR 3%	NA
	WP1066	JAK2, STAT3	35575067	Phase I	Recurrent	GBM *n* = 8	PFS 2.3m (95% CI 1.7–NA) OS form initial diagnosis 25m (95% CI 22.5–NA)	PD 100%	NA
Tumourcell + angiogenesis	Axitinib	VEGFR1, VEGFR2, VEGFR3 PDGFRß, c-Kit	28988341	Phase II	Recurrent	GBM *n* = 50	PFS 12.4w (95% CI 11–13) OS 29w (95% CI 20–38)	NA	MGMT-promoter hypermethylation is significantly correlated with PFS and OS
			26935577	Phase II	Recurrent	GBM *n* = 22	PFS 13w (95% CI 11–14) OS 29w (95% CI, 17–40)	CR 9%, PR 18%, SD 14%	No difference in PFS or OS for tested mutations.
			33067319	Phase II	Recurrent	GBM *n* = 27	PFS6 18.5% OS 18w	ORR 22.2%, CR 3.7%, PR 18.5%, SD 25.9%, PD 51.9%	NA
	Cabozantinib	VEGFR1-3, c-Met, Ret, Kit, Flt-1/3/4, Tie2, AXL	29016998	Phase II	Recurrent	GBM *n* = 152	PFS 3.7m OS 7.7–10.4m (depending on dose)	140 mg/day: ORR 17.6%, PR 17.6%, SD 58.8%, PD 11.8% 100 mg/day: ORR 14.5%, PR 14.5%, SD 67.5%, PD 12.0%	NA
			29036345	Phase II	Recurrent	GBM *n* = 70	PFS 2.3m OS 4.6m	140 mg/day: ORR 8.3%, PR 8.3%, SD 50.0%, PD 16.7% 100 mg/day: ORR 3.4%, PR 3.4%, SD 46.6%, PD 27.6%	NA
	Dovitinib	FLT3/c-Kit, FGFR1/3, VEGFR1-4, PDGF	31292802	Phase II	Recurrent	GBM *n* = 19	TTP 1.8m (95% CI 1.4–1.8) OS 5.6m (95% CI 4.2–8.1)	NA	No impact on OS. Higher BMP 9, CD73, endoglin and VEGF D, and lower TSP 2 were associated with poorer PFS
			27100354	Phase I	Recurrent	GBM *n* = 12	PFS 1.8m (95% CI 1.7–1.9) OS 9.5m (95% CI 2.6–16.4)	CR 0%, PR 0%, SD 36.4%, PD 63.4%	Presence of FGFR-TACC gene fusion did not affect PFS-6
	Infigratinib	FGFR 1/2/3	35344029	Phase II	Recurrent	GBM *n* = 19 AA *n* = 5 Other *n* = 2	PFS 1.7m (95% CI 1.1–2.8) OS 6.7m (95% CI 4.2–11.7)	ORR 4.8%, PR 4.8%, SD 28.6%, PD 61.9%	Tumours harbouring FGFR1 or FGFR3 point mutations or FGFR3-TACC3 fusions showed durable disease control for more than 1 year
	Nintedanib	VEGFR1/2/3, FGFR1/2/3 and PDGFRα/β	23184145	Phase II	Recurrent	GBM *n* = 25	PFS-6 4.0% (95% CI 0.1–20.4) OS 6m (95% CI 3.6–8.4)	CR 0%, PR 0%, SD 12.0%, PD 88.0%	NA
			25338318	Phase II	Recurrent	GBM *n* = 22	Bevacizumab-naive: PFS 28d (95% CI 27–83) OS 6.9m (95% CI 3.7–8.1). bevacizumab-treated: PFS 28d (95% CI 22–28) OS 2.6m (95% CI 1.0–6.9)	Bevacizumab-naive: CR 0%, PR 0%, SD 33%, PD 67% Bevacizumab-treated: CR 0%, PR 0%, SD 10%, PD 90%	NA
	Regorafenib	VEGFR1, VEGFR2, VEGFR3, PDGFRa, PDGFRβ, Kit (c-Kit), RET (c-RET) and Raf-1, FGFR1, FGFR2, Abl	30522967	Phase II	Recurrent	GBM *n* = 59	PFS 2.0m (95% CI 1.9–3.6) OS 7.4m (95% CI 5.8–12.0)	CR 2%, PR 3%, SD 39%, PD 56%	NA
	Sunitinib	VEGFR1, VEGFR2 VEGFR3, PDGFRa, PDGFRß, c-Kit, FLT3, CSF-1R, RET	22832897	Phase II	Recurrent	GBM *n* = 16	PFS 1.4m (95% CI 1.2–4.8) OS 12.6m (95% CI 3.9–18.1)	CR 0%, PR 0%, SD 31.3%	NA
			23086433	Phase II	Recurrent	GBM *n* = 31	PFS 1.08m (95% CI 0.92–2.47) OS 9.4m (95% CI 6.15–21.90)	Rate of radiographic response 10%, Levin 23%	NA
			24311637	Phase II	Recurrent	GBM *n* = 40	PFS 2.2m (95% CI 1.8–3.3) OS 9.2m (95% CI 11.9–49.2)	ORR 0%, SD 12.5%, PD 82.5%	c-KIT expression in vascular endothelial cells was associated with improved PFS (2.3m) versus c-KIT negative vascular endothelial cells (1.7m) (*p* = 0.025). No or low expression of PDGFR-α in tumour cells was associated with improved PFS (*p* = 0.043) but not with OS.
			24424564	Phase II	De novo	GBM *n* = 12	PFS 7.7w (95% CI 7.2–8.2) OS 12.8w (95% CI 0.5–23.8)	ORR 0%, SD 8.3%, PD 91.7%	NA
	SYHA1813	VEGFR, CSF1R	36884148	Phase I	Recurrent	GBM *n* = 4	NA	PR 25%	NA
	Tivozanib	VEGFR1/2/3, PDGFR, c-Kit Low activity against FGFR-1, Flt3, c-Met, EGFR and IGF-1R	27853960	Phase II	Recurrent	GBM *n* = 10	PFS 2.3m (95% CI 1.5–4.0) OS 8.1m (95% CI 5.2–12.5)	CR 10%, PR 10%, SD 40%, PD 40%	None of the investigated blood biomarkers were associated with OS or PFS.
Tumourcell + micro-environment	AEE788	EGFR, HER2/ErbB2, VEGFR2/KDR, c-Abl, c-Src, Flt-1	22392572	Phase I	Recurrent	GBM *n* = 64	PFS 2.7m (90% CI 1.9–2.8)	CR 0%, PR 0% SD 17%	p-KDR was a significant predictor of PFS (*p* = 0.01)
	Bosutinib	Src/Abl, PI3K/AKT/mTOR, MAPK/ERK, JAK/STAT3. Lyn, HCK. Promotes autophagy	25411098	Phase II	Recurrent	GBM *n* = 9	PFS 7.71w (95% CI 2.6–7.9) OS 50w (95% CI 2.9–NA)	PD 100%	NA
	Cediranib	VEGFR(KDR), Flt1/4, c-Kit, PDGFRβ, induces autophagic vacuole accumulation.	20458050	Phase II	Recurrent	GBM *n* = 30	PFS 117d (95% CI 82–145) OS 227d (95% CI 177–293)	2D measurements: PR 26.6% 3D measurements: PR 56.7%	At baseline, no biomarkers showed correlations with PFS or OS
			23940216	Phase III	Recurrent	GBM *n* = 131	PFS 92d OS 8m	CR 0,8%, PR 14.4%, SD 64.4%, PD 8.5%	Baseline VEGF levels did not have a significant effect on PFS or OS
	Diazepinomicin (TLN-4601)	RAS, peripheral benzodiazepine receptor (PBR)	22048878	Phase II	Recurrent	GBM *n* = 17	PFS-6 0% OS 150d	CR + PR + SD 21.4%	NA
	Erdafitinib (JNJ-42756493)	FGFR1/2/3/4, RET (c-RET), CSF-1R, PDGFR-α/PDGFR-β, FLT4, Kit (c-Kit), VEGFR-2	26324363	Phase I	Recurrent	GBM *n* = 3	NA	PR 66.7%	NA
	Imatinib	v-Abl, c-Kit, SCF, and PDGFR, induces autophagy	18824712	Phase II	Recurrent	GBM *n* = 51	PFS 1.8m (95% CI 1.7–2.3) OS 5.9m (95% CI 4.2–7.8)	PR 6%, SD 26%	PFS was not correlated with PDGFRα SNPs.
			16914578	Phase I+II	Recurrent	Phase I: GBM *n* = 35 Phase II: GBM *n* = 24	PFS6 3%	Phase I: CR 0%, PR 2.9%, SD 34.3% Phase II: CR 0%, PR 5,9%, SD 17.6%	NA
			31514200	Phase II	De novo and recurrent	De novo: GBM *n* = 19 Recurrent: GBM *n* = 32	De novo: PFS 2.8m (95% CI 0.3–8) OS 5.0m (95% CI 0.8–30) Recurrent: PFS 2.1m (95% CI 0.3–19.3) OS 6.5m (95% CI 0.3–51.5)	NA	NA
			19789313	Phase II	De novo	GBM *n* = 20	OS 6.2m	CR 0%, PR 0%, SD 90%, PD 5%, nonevaluable 5%	NA
	ONC201	Akt and ERK to induce TNF-related apoptosis-inducing ligand (TRAIL)	31702782	Phase II	Recurrent	GBM *n* = 20	PFS 1.8m OS 7.5m	CR 0%	NA
	Pazopanib	VEGFR1, VEGFR2, VEGFR3, PDGFRa/β,c-Kit, FGFR1, c-Fms Induces autophagy	20200024	Phase II	Recurrent	GBM *n* = 35	PFS 12w (95% CI 8–14) OS 35w (95% CI 24–47)	ORR 5.9% (95% CI: 0.7–21%). SD 59%, PD 32%	NA
	Pexidartinib (PLX3397)	CSF-1R, Kit (c-Kit), FLT3, PDGFRβ	26449250	Phase II	Recurrent	GBM *n* = 37	PFS-6 8.8% (90% CI 3.5%, 21.6%) OS 9.4m (90% CI 6.67–NA)	CR 0%, PR 0%	PDGFRA amplification and gains did not correlate significantly with PFS6 or other parameters.
	Ponatinib	Abl, PDGFRα, VEGFR2, FGFR1, Src	31444999	Phase II	Recurrent	GBM *n* = 15, Bevacizumab-refractory	PFS 28d (95% CI 27–30) OS 98d (95% CI 56–257)	SD 13.7%, PD 66.7%	NA
	Temsirolimus	mTOR, induces autophagy	16012795	Phase II	Recurrent	GBM *n* = 43	PFS 9w	PR 4.7%, SD 46.5%	NA
			15998902	Phase II	Recurrent	GBM *n* = 65	TTP 2.3m (95% CI 1.9–3.2) OS 4.4m (95% CI 3.6–4.8)	ORR 0%	Significant association between neuroimaging response and p70s6 kinase phosphorylation in baseline tumour samples (*p* = 0.04)
	Vandetanib	VEGFR2, VEGFR3, EGFR, RET, induces autophagy by increasing the level of reactive oxygen species (ROS)	23099652	Phase I/II	Recurrent	GBM *n* = 32	PFS 1.3m (95% CI 0.9–1.9) OS 6.3m (95% CI 3.8–8.5)	ORR 15%, CR 3.7%	NA
	Vemurafenib	B-RafV600E, induces cell autophagy	30351999	Phase II	Recurrent	GBM *n* = 6, BRAFV600mt AA *n* = 5, BRAFV600mt	PFS 5.3m (95% CI 1.8–12.9), OS 11.9m (95% CI 8.3–40.1)	CR 0%, PR 9.1%, SD 45.5%, PD 27.3%	NA
Stemcell	RO4929097	y-secretase, Aβ40 and Notch	33027815	Phase II	Recurrent	GBM *n* = 47	PFS 1.7m (95% CI 1.2–1.8) OS 7.0m (95% CI 5.4–9.1)	CR 2%, PR 0%, SD 6%, PD 81%	NA
			NCT01122901	Phase II	Recurrent	GBM *n* = 40	PFS 1.7m (95% CI 1.1–1.8) OS 6.6m (95% CI 5.3–10.5)	CR 2.5%, PR 0%, SD 7.5%, PD 82.5%	NA

GBM: glioblastoma, AO: anaplastic oligodendroglioma, AA: anaplastic astrocytoma, AG: anaplastic glioma, MG: malignant glioma, PFS: progression-free survival, TTP: time until progression, OS: overall survival, CR: complete response, PR: partial response, SD: stable disease, PD: progressive disease, DCR: disease control rate, ORR: objective response rate. m: months, w: weeks and d: days.

**Table 3 cancers-16-03021-t003:** Clinical trials investigating the combination of small molecule inhibitors with the standard of care.

Compounds	Targets Specified	PMID	Trial	De Novo/ Recurrent	Study Population	PFS/OS	Response Rate	Biomarker Analysis
Afatinib + TMZ	Afatinib: EGFR, HER2, HER4	25140039	Phase II	Recurrent	GBM *n* = 39	PFS 1.53m OS 8.0m	CR 2.6%, PR 5.1%, SD 35.9%, PD 43.6%	No statistically significant relation between EGFRvIII and treatment outcome
Anlotinib + TMZ	VEGFR2/3, FGFR1-4, PDGFR α/β, c-Kit, and Ret	37477938	Phase II	Recurrent	GBM *n* = 21	PFS 7.3m (95% CI 4.9–9.7) OS 16.9m (95% CI 7.8–26.0)	ORR 81% (95% CI 62.6–99.3), CR 43%, PR 38%	NA
Bortezomib + bevacizumab	Bortezomib: 20S proteasome, inhibits NF-κB and induces ERK phosphorylation to suppress cathepsin B and inhibit the catalytic process of autophagy	NCT00611325	Phase II	Recurrent	GBM *n* = 56	EAIED: PFS 2m (95% CI 2–4) OS 8m (95% CI 5–11) Non-EAIED: PFS 2.5m (95% CI 1–4) OS 6m (95% CI 4–10)	EAIED: RRR 7.1% (95% CI 0–16.6) Non-EAIED: RRR 39.3% (95% CI 21.2–57.4)	NA
Bortezomib + TMZ	Bortezomib: 20S proteasome, inhibits NF-κB and induces ERK phosphorylation to suppress cathepsin B and inhibit the catalytic process of autophagy	27300524	Phase II	Recurrent	GBM *n* = 9 AO grade III *n* = 1	PFS 2.6m OS 8.9m	NA	NA
		32578964	Phase Ib	Recurrent	GBM *n* = 10	OS 21.4m	NA	NA
Buparlisib + bevacizumab	Buparlisib: PI3K	31392595	Phase II	Recurrent	GBM *n* = 76	PFS 4.0m (95% CI 3.4–5.4) Bevacizumab-naïve: OS 10.8m (95% CI 9.2–13.5) Bevacizumab treated: OS 6.6m (95% CI 4.0–14.6)	ORR 26%. CR 11%, PR 16%, SD 33%, PD 29%	PTEN and PIK3CA did not affect the treatment response.
Buparlisib + lomustine	Buparlisib: PI3K	32665311	Phase Ib/II	Recurrent	GBM *n* = 18	NA	CR 0%, PR 0%, SD 11.1%, PD 77.8%	NA
CT-322 + irinotectan	CT-322: VEGFR-2	25388940	Phase II	Recurrent	GBM *n* = 63	PFS 8.8m	ORR 0%	NA
Dasatinib + bevacizumab	Dasatinib: Abl, Src, c-Kit, induces autophagy	31290996	Phase II	Recurrent	GBM *n* = 83	PFS 3.2m OS 7.3m	ORR 15.7%, SD 57.8%	No associations between VEGFR2, Y416.SRC (pSRC), CD31, LYN and YES and PFS or OS.
Enzastaurin + bevacizumab	Enzastaurin: PKCβ, PKCα, PKCγ and PKCε	26643807	Phase II	De novo	GBM *n* = 37	PFS 2.0m OS = 7.5m	ORR 22%, SD 54%	No correlation with treatment response and p-GSK-3 levels.
Erlotinib + bevacizumab	Erlotinib: EGFR	23132371	NA	Recurrent	GBM *n* = 4	PFS 10.5m OS 17.0m	Response rate 100%	NA
		20716591	Phase II	Recurrent	GBM *n* = 25	PFS 18w (95% CI 12.0–23.9) OS 44.6w (95% CI28.4–68.7)	CR 4%, PR 46%, SD 42%, PD 8%	Patients with positive pS6 had a 3.4 times greater risk of progression compared with patients with negative pS6 (*p* = 0.05). Patients with lower values for VEGFR-2 were more likely to survive more than 1 year than those with high values of VEGFR-2 (*p* = 0.0079)
		26476729	Phase II	De novo after treatment RT and TMZ, no progression	GBM *n* = 46	PFS 9.2m (95% CI 6.4–11.3) OS 13.2m (95% CI 10.8–19.6)	CR 8.7%, PR 26.1%, SD 60.9%, PD 0%, 4.3% unknown	NA
Erlotinib + TMZ	Erlotinib: EGFR	16443950	Phase I	Stable or recurrent	GBM *n*=39	NA	PR 2.6%	NA
Everolimus + TMZ	Everolimus: mTOR inhibitor of FKBP12, autophagy	22160854	Phase I	De novo	GBM *n* = 17, non-EAIED GBM *n* = 11, EAIED	NA	Non-EIAED: ORR 17.6% (95% CI: 3.8–43.4%). CR 0%, PR 3/17, SD 9/17, PD 5/17 EIAED: CR 0%, PR 0%, SD 7/11, PD 4/11,	No differences in response and survival between patients with PTEN intact and deleted tumours.
Imatinib + hydroxyurea	Imatinib: v-Abl, c-Kit and PDGFR, induces autophagy.	16361636	Phase II	Recurrent	GBM *n* = 33	PFS 14.4w (95% CI 8.3–16.6) OS 48.9w (95% CI 25.7–71.1)	CR 3%, PR 6%, SD 42%, PD 48%	NA
		19904263	Phase II	Recurrent	GBM *n* = 231	PFS 5.6w (95% CI 4.1–7.9) OS 26w (95% CI 21.3–31.3)	CR 0.4%, PR 3.0%, SD 19.5%, PD 61.5%	Patients with increased c-KIT had significant longer PFS.
		19688297	Phase III	Recurrent	GBM *n* = 120	PFS 6w(95% CI 6–7) OS 21w	PD 40%	NA
Imatinib mesylaat + TMZ	Imatinib: v-Abl, c-Kit and PDGFR, induces autophagy.	18359865	Phase I	Stable & recurrent	GBM *n* = 52	PFS 26.6w (95% CI 9.9–36.4) OS 45.1w (95% CI 36.1–59.1)	CR 0%, PR 12%, SD 42%	NA
Lapatinib + TMZ	Lapatinib: EGFR, ERBB2	23292205	Phase I	Recurrent	GBM *n* = 14 AA *n* = 2	PFS 2.4m OS 5.9m	CR 0%, PR 6.3%, SD 31.3%	NA
Lonafarnib + TMZ	FPTase inhibitor for H-ras, K-ras-4B, N-ras	23633392	Phase I/Ib	Recurrent	GBM *n* = 35	PFS 3.9m (95% CI 2.5–8.4) OS 13.7m (95% CI 8.9–22.1)	CR 5.9%, PR 17.6%, SD 47.1%, PD 29.4%	NA
Olaparib + TMZ	Olaparib: PARP1, PARP2	32347934	Phase I	Recurrent	GBM *n* = 36	PFS6 39% (95% CI: 23.1–56.5%)	NA	NA
Panobinostat + bevacizumab	Panobinostat: HDAC, autophagy	25572329	Phase II	Recurrent	GBM *n* = 24	PFS 5m (95% CI 3–9) OS 9m (95% CI 6–19)	CR 0%, PR 29.2%, SD 58.3%, PD 12.5%	NA
Sorafenib + bevacizumab	Sorafenib: Raf-1, B-Raf, VEGFR-2, VEGFR-3, PDGFR-β, Flt-3, c-KIT	23833308	Phase II	Recurrent	GBM *n* = 54	PFS 2.9m (95% CI 2.3–3.6) OS 5.6m (95% CI 4.7–8.2)	ORR 18.5%, SD 63%	PFS6 success was increased for VEGFR promoter mutant rs699947 and rs833061 and PFS6 success decreased for mutant rs1005230 and rs1570360. PFS6 success was increased for VEGFR2 promoter heterozygous rs2071559.
Sorafenib + TMZ	Sorafenib: Raf-1, B-Raf, VEGFR-2, VEGFR-3, PDGFR-β, Flt-3, c-KIT	20443129	Phase II	Recurrent	GBM *n* = 32	PFS 6.4w (95% CI 3.9–11.7) OS 41.5w (95% CI 24.1–55.1)	CR 0%, PR 3%, SD 47%, PD 50%	NA
		23898124	Phase II	Recurrent	GBM *n* = 43	TTP 3.2m (95% CI 1.8–4.8) OS 7.4m (95% CI 5.6–9.0)	CR 0%, PR 12%, SD 43%, PD 48%	NA
Sunitinib + irinotecan	Sunitinib: VEGFR2, PDGFRß, c-Kit, IRE1α	21744079	Phase I	Recurrent,	GBM *n* = 15 MG grade III *n* = 10	PFS 6.9w (95% CI 5.7–17.7), OS 53.1w (95% CI 30.3–87.9)	CR 0%, PR 4%, SD 36%, PD 60%	NA
		27680966	Phase II	Recurrent	GBM *n* = 6	PFS-6 not reached	ORR 17%	NA
Tandutinib + bevacizumab	Tandutinib: FLT3, c-Kit, PDGFR	26860632	Phase II	Recurrent	GBM *n* = 37	PFS 4.1m OS 11m	PR 24%	NA
Temsirolimus + bevacizumab	Temsirolimus: mTOR, induces autophagy	23564811	Phase II	Recurrent	GBM *n* = 10	PFS 8w OS 15w	CR 0%, PR 0%, SD 20%	NA
Trebananib + bevacizumab	Trebananib: Angiopoietin 1 and angiopoietin 2 blocking peptibody	29266174	Phase II	Recurrent	GBM *n* = 37	PFS 3.6m (95% CI 1.9–5.5) OS 9.5m (95% CI 7.5–14.7)	ORR 27%, CR 0%, PR 27%, SD 41%	High plasma VEGF was associated with poor PFS and OS. High plasma IL-8 was associated with shorter OS (*p* < 0.05)
		32154928	Phase II	Recurrent	GBM *n* = 57	PFS 4.2m (95% CI 3.7–5.6)	CR 0%, PR 4.2%, SD 18.8% PD, 77.1%	NA
		NCT01290263	Phase I/II	Recurrent	GBM *n* = 37	PFS 108d OS 285d	CR 0%, PR 10.8%, SD 54%, PD 27%	NA
Velparib + TMZ	Velparib: PARP1, PARP2, autophagy	26508094	Phase I/II	Recurrent	GBM *n* = 146, bevacizumab naïve GBM *n* = 69, bevacizumab failure	BEV-naïve: OS 10.3m low TMZ dose OS 10.7m high TMZ dose BEV failure: OS 4.7m low TMZ dose OS 4.7m high TMZ dose	BEV-naïve CR 1.9%, PR 1.9%, SD 1.9% BEV failure: CR 5.3%, PR 0%	NA
Vorinostat + bevacizumab	Vorinostat: HDAC	29133513	Phase II	Recurrent	GBM *n* = 40	PFS 3.7m (95% CI 2.9–4.8) OS 10.4m (95% CI 7.6–12.8)	RRR 22.5%, CR 0%, PR 22.5%	NA
		32166308	Phase II	Recurrent	GBM *n* = 44	PFS 3.68m (95% CI 2.33–3.94) OS 7.79m (95% CI 5.06–9.63)	NA	NA
	Vornistat + bevacizumab + TMZ	29264836	Phase I/II	Recurrent	GBM *n* = 39	PFS 6.7m (95% CI 4.8–9.4) OS 12.5m (95% CI 8.8–14.3)	RRR 56% (95% CI 41–71)	NA

GBM: glioblastoma, AO: anaplastic oligodendroglioma, AA: anaplastic astrocytoma, AG: anaplastic glioma, MG: malignant glioma, PFS: progression-free survival, TTP: time until progression, OS: overall survival, CR: complete response, PR: partial response, SD: stable disease, PD: progressive disease, DCR: disease control rate, ORR: objective response rate, RRR: radiographic response rate. EIAED: enzyme-inducing antiepileptic drug. m: months, w: weeks and d: days.

**Table 4 cancers-16-03021-t004:** Clinical trials investigating combinations of small molecule inhibitors.

Compounds	Targets Specified	PMID	Trial	De Novo/ Recurrent	Study Population	PFS/OS	Response Rate	Biomarker Analysis
Cediranib + cilengitide	Cediranib: VEGFR(KDR), Flt1/4, c-Kit, PDGFRβ, induces autophagic vacuole accumulation. Cilengitide: Integrins ανβ3 and ανβ5	26008604	Phase I	Recurrent	GBM *n* = 45	PFS 1.9m (95% CI 1.5–2.8) OS 6.5m (95% CI 5.2–7.6)	CR 4.4%, PR 4.4%, SD 28.9%, PD 46.7%	NA
Cediranib + gefitinib	Cediranib: VEGFR(KDR), Flt1/4, c-Kit, PDGFRβ, induces autophagic vacuole accumulation. Gefitinib: EGFR	27232884	Phase II	Recurrent	GBM *n* = 19	PFS 3.6m OS 7.2m	CR 0%, PR 42%	NA
Erlotinib + sirolimus	Erlotinib: EGFR Sirolimus: mTOR	19562254	Phase II	Recurrent	GBM *n* = 32	PFS 6.9w (95% CI 3.9–11) OS 33.8w (95% CI 21.9–53.6)	SD 27%, CR 0%, PR 0%	No association between EGFR, PTEN, EGFRvIII, pS6 and pMAPK PFS6, borderline significance with p-AKT (*p* = 0.045).
Erlotinib + sorafenib	Erlotinib: EGFR Sorafenib: Raf-1, B-Raf, VEGFR-2, VEGFR-3, PDGFR-β, Flt-3, c-KIT	23328813	Phase II	Recurrent	GBM *n* = 56	PFS 2.5m (95% CI 1.8–3.7) OS 5.7m (95% CI 4.5–7.9)	CR 0%, PR 5% (all unconfirmed), SD 41%, PD 45%	NA
		33235994	Phase I/II	Recurrent	GBM *n* = 19	PFS 1.8m OS 5m	CR 0%, PR 0%	NA
Erlotinib + temsirolimus	Erlotinib: EGFR Temsirolimus: mTOR, induces autophagy	24470557	Phase I/II	Recurrent	GBM *n* = 42	PFS 8w (95% CI 8–10)	CR 0%, PR 0%, SD 29%	No significant correlation between EGFRvIII, EGFR amplification PTEN, p-AKT or pS6^S235/236^ and PFS.
Gefitinib + sirolimus	Sirolimus: mTOR Gefitinib: EGFR	16467100	Phase I	Recurrent	GBM *n* = 29 AG *n* = 5	PFS 8.2w (95% CI 7.5–18.6)	PR 5.9%, SD 41%	NA
Pazopanib + lapatinib	Pazopanib: EGFR1, VEGFR2, VEGFR3, PDGFR, FGFR, c-Kit, c-Fms/CSF1R, cathepsin B activation, autophagy. Lapatinib: EGFR, ERBB2	23363814	Phase I/II	Recurrent	GBM *n* = 19, biomarker positive (PTEN and/or EGFRvIII) GBM *n* = 22, biomarker negative	Biomarker positive: PFS 56d (95% CI 45–113) Biomarker negative: PFS 62d (95% CI 56–90)	Overall: CR 0%, PR 5%, SD 34%, PD 61% Biomarker positive: CR 0%, PR 5%, SD 37%, PD 58% Biomarker negative: CR 0%, PR 5%, SD 32%, PD 64%	NA
Temsirolimus + perifosine	Temsirolimus: mTOR, induces autophagy Perifosine: Akt	32293798	Phase I	Recurrent	GBM *n* = 17 Other MG *n* = 19	PFS 2.7m (95%CI 1.8–9.2) OS 10.4m (95% CI 7.2–16.7)	PR 3.4%, SD 44.8%, PD 51.7%	NA
Temsirolimus + sorafenib	Temsirolimus: mTOR, induces autophagy Sorafenib: Raf-1, B-Raf, VEGFR-2, VEGFR-3, PDGFR-β, Flt-3, c-KIT	29313954	Phase I + II	Recurrent	Arm B (anti-VEGF therapy naïve): GBM *n* = 49 Arm D (prior anti-VEGF therapy): GBM *n* = 44	Arm B: PFS 2.7m OS 6.6m Arm D: PFS 1.9m OS 3.9m	Arm B: PR 2.4%, SD 64%, PD 27% Arm D: SD 46%, PD 51%	NA
		23099651	Phase I/II	Recurrent	GBM *n* = 18	PFS 8w (95% CI 5–9)	CR 0%, PR 11.8%	NA
Tipifarnib + sorafenib	Tipifarnib: FTase Sorafenib: Raf-1, B-Raf, VEGFR-2, VEGFR-3, PDGFR-β, Flt-3, c-KIT	28988377	Phase I	Recurrent	GBM *n* = 24	PFS 55d OS 4.38m	NA	NA
Trametinib + dabrafenib	Trametinib: MEK 1/2, activates autophagy Dabrafenib: BRAFV600	PMC6217670	Phase II	Recurrent	HGG *n* = 45, BRAFV600-mutant	PFS 1.9m (95% CI 1.7–18.5) OS 11.7m (95% CI 6.4–not reached)	NA	NA
		34838156	Phase II	Recurrent	GBM *n* = 31, BRAFV600E-mutant	PFS 2.8Mm OS 13.7m	ORR 32%, PD 45%, CR 6%, PR 26%	NA
Vandetanib + sirolimus	Vandetanib: VEGFR2, VEGFR3, EGFR, induces autophagy by increasing the level of reactive oxygen species (ROS) Sirolimus: mTOR	25503302	Phase I	Recurrent	GBM *n* = 19	PFS 2.1m (95% CI 0.9–3.1) OS 7.7m (95% CI 4.7–9.3)	PR 10.5%	NA
Vatalanib + imatinib + hydroxyurea	Imatinib: v-Abl, c-Kit and PDGFR, induces autophagy. Vatalanib: VEGFR2/KDR, VEGFR1/Flt-1, VEGFR3/Flt-4.	19248046	Phase I	Recurrent	GBM *n* = 34 MG grade III *n* = 3	PFS 12w OS 48w	PR 24%, SD 49%, PD 27%	NA
Vorinostat + bortezomib	Vorinostat: HDAC Bortezomib: 20S proteasome, inhibits NF-κB and induces ERK phosphorylation to suppress cathepsin B and inhibit the catalytic process of autophagy	22090453	Phase II	Recurrent	GBM *n* = 34	TTP 1.5m OS 3.2m	Partial objective response 2.7%	NA

GBM: glioblastoma, AO: anaplastic oligodendroglioma, AA: anaplastic astrocytoma, AG: anaplastic glioma, HGG: high grade glioma, MG: malignant glioma. PFS: progression-free survival, TTP: time until progression, OS: overall survival, CR: complete response, PR: partial response, SD: stable disease, PD: progressive disease, DCR: disease control rate, ORR: objective response rate, RRR: radiographic response rate. m: months, w: weeks and d: days.

## Data Availability

No new data were created or analysed in this study. Data sharing is not applicable to this article.

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
