# Peer review of "Opportunities and Challenges of Small Molecule Inhibitors in Glioblastoma Treatment: Lessons Learned from Clinical Trials"

_cancers, 2024, doi:10.3390/cancers16173021_

Round 1

Reviewer 1 Report

Comments and Suggestions for Authors

Hoosemans et al provide a review on small molecule inhibitors in the treatment of glioblastoma. This provided a comprehensive review of clinical trials involving a variety of single target and multi-target small molecules as monotherapy, in combination with chemotherapy/bevacizumab and in combination with other small molecule inhibitors. Overall, this was a well-written and comprehensive summary of clinical trials performed on small molecule inhibitors that was informative and importantly, easy to read and understand and requires no modification. 

With regard to individual sections: 

SIMPLE SUMMARY AND ABSTRACT: Provided appropriate and accurate summaries of the information provided in the text.

INTRODUCTION: Provides a brief background to the treatment of glioblastoma and the rationale for  use of small molecule inhibitors in its treatment of glioblastoma, as well as the potential barriers to successful therapy.

SECTION 2 CLINICAL TRIALS WITH SMALL MOLECULE INHIBITORS IN GBM: This has separate sections on clinical trials of monotherapy with single target and multitarget small molecule inhibitors, clinical trials of small molecule inhibitors combined with chemotherapy/bevacizumab, and clinical trials with combinations of small molecule inhibitors. This is accompanied by four separate tables, one for each topic, that supplement the narrative text without significant redundancy.

DISCUSSION: This was divided into three sections, intrinsic versus acquired resistance, influence of tumor microenvironment and clinical trial design. The discussion addressed the reasons for failure of most clinical trials with a comprehensive and understandable rationale and used the information discussed to consider future clinical trial design.

SECTION 4, CONCLUSION AND FUTURE PERSPECTIVES: This section adequately summarize the content of the submission.

REFERENCES: All references were up-to-date and current and pertinent to the subject matter.

TABLES AND FIGURES: There were no figures. All tables adequately supplemented the narrative text with regard to provision of information regarding clinical trials previously performed and their outcomes.

Author Response

Comment 1: 

Hoosemans et al provide a review on small molecule inhibitors in the treatment of glioblastoma. This provided a comprehensive review of clinical trials involving a variety of single target and multi-target small molecules as monotherapy, in combination with chemotherapy/bevacizumab and in combination with other small molecule inhibitors. Overall, this was a well-written and comprehensive summary of clinical trials performed on small molecule inhibitors that was informative and importantly, easy to read and understand and requires no modification. 

With regard to individual sections: 

SIMPLE SUMMARY AND ABSTRACT: Provided appropriate and accurate summaries of the information provided in the text.

INTRODUCTION: Provides a brief background to the treatment of glioblastoma and the rationale for  use of small molecule inhibitors in its treatment of glioblastoma, as well as the potential barriers to successful therapy.

SECTION 2 CLINICAL TRIALS WITH SMALL MOLECULE INHIBITORS IN GBM: This has separate sections on clinical trials of monotherapy with single target and multitarget small molecule inhibitors, clinical trials of small molecule inhibitors combined with chemotherapy/bevacizumab, and clinical trials with combinations of small molecule inhibitors. This is accompanied by four separate tables, one for each topic, that supplement the narrative text without significant redundancy.

DISCUSSION: This was divided into three sections, intrinsic versus acquired resistance, influence of tumor microenvironment and clinical trial design. The discussion addressed the reasons for failure of most clinical trials with a comprehensive and understandable rationale and used the information discussed to consider future clinical trial design.

SECTION 4, CONCLUSION AND FUTURE PERSPECTIVES: This section adequately summarize the content of the submission.

REFERENCES: All references were up-to-date and current and pertinent to the subject matter.

TABLES AND FIGURES: There were no figures. All tables adequately supplemented the narrative text with regard to provision of information regarding clinical trials previously performed and their outcomes.

Response 1: Thank you for your kind words. 

Reviewer 2 Report

Comments and Suggestions for Authors

These are the comments which can improve the paper:

1. I would like to suggest that the authors come up with a diagram of some sort to describe the  current pipeline of drugs which are currently being developed for glioblastoma.

2. I suggest that some additional aspects of therapeutic and molecular mechanisms of the current drugs and drugs on their way to the market should be described in this article.

Author Response

Comments 1: I would like to suggest that the authors come up with a diagram of some sort to describe the  current pipeline of drugs which are currently being developed for glioblastoma.
Response 1: Thank you for your suggestion. Different small molecule inhibitors have been developed in the past decade against different oncogenic markers in different types of cancer. However, the current pipeline of drugs contains no small molecule inhibitors that have shown a signal of response in GBM. No small molecule inhibitors are expected to become standard of care in GBM in the near future. 

Comments 2:  I suggest that some additional aspects of therapeutic and molecular mechanisms of the current drugs and drugs on their way to the market should be described in this article.
Response 2: Thank you for your comment. The therapeutic and molecular mechanisms of small molecule inhibitors are determined by the specific genetic drivers they target. In this review, we outline the therapeutic mechanisms of these inhibitors in the introduction. Furthermore, the targets of all small molecule inhibitors are detailed in the tables. Additionally, we emphasize the significance of exploring newly developed small molecule inhibitors, such as paxalisib, that exhibit enhanced blood-brain barrier (BBB) permeability. We understand the important comment of the reviewer, however since we have synthesized and reviewed all the most important aspects of the different compounds with references, we have decided not to add additional details, also in favour of the clarity of the review. 

Reviewer 3 Report

Comments and Suggestions for Authors

The authors have described the role of small molecule inhibitors in Glioblastoma treatment. However following points needed to be addressed before further process:

1. What are the main reasons for the failure of clinical trials with small molecule inhibitors in glioblastoma treatment?

2. Many trials are biomarker-naïve and unable to identify ‘on-target’ effects of specific drugs, monitor treatment response, or identify resistance mechanisms. This challenge arises because re-sampling of the tumor during treatment is not feasible due to its location and the unexplored potential of liquid biopsy in GBM.  Authors should add data to this.

Comments on the Quality of English Language

Can be improved

Author Response

Comments 1: What are the main reasons for the failure of clinical trials with small molecule inhibitors in glioblastoma treatment?
Response 1: Thank you for your question. The main reason for failure of clinical trials with small molecule inhibitors in GBM is the presence of intra- and intertumoral heterogeneity, which precludes the development of a universally effective treatment for these patients [1-3]. We have rephrased this sentence, which can be found on page 2, line 50.
Additionally, GBM tumours can circumvent the therapeutic effects of these drugs by activating alternative signalling pathways [4-6]. Other factors contributing to the disappointing results of clinical trials in GBM include suboptimal drug concentrations at the tumour site due to the blood-brain barrier (BBB) and inadequate patient stratification in clinical trials because of a lack of a biomarker, which fails to account for intertumoral heterogeneity [7,8]. These aspects are discussed in greater detail in the introduction and discussion sections of this manuscript.

Comments 2: Many trials are biomarker-naïve and unable to identify ‘on-target’ effects of specific drugs, monitor treatment response, or identify resistance mechanisms. This challenge arises because re-sampling of the tumor during treatment is not feasible due to its location and the unexplored potential of liquid biopsy in GBM.  Authors should add data to this.
Response 2: Thank you for your important comment. In a comprehensive study involving a large cohort of matched primary and recurrent glioblastoma (GBM) samples, French et al. demonstrated that the mutational status of driver genes exhibits a high degree of stability [9]. The most precise and sensitive method for detecting 'on-target' effects of small molecule inhibitors or mechanisms of resistance remains the analysis of biopsy material. However, the biopsy procedure carries risks, including potential haemorrhages, infections and damage of eloquent brain areas with subsequent loss of function, leading to a preference for less invasive approaches.

Various types of liquid biopsies have been explored in the context of GBM; however, no clinically validated GBM-specific biomarker has yet been identified. Liquid biopsies face several challenges, including the presence of biomarkers in extremely low concentrations in the blood and their short half-life, which complicates detection. Additionally, the sample obtained may not fully represent the entire tumour due to intratumoral heterogeneity and most aggressive tumour cells. So far, no prognostic or predictive biomarkers have been identified in the blood of GBM patients [10]. Consequently, further studies are required to evaluate the sensitivity and specificity of liquid biopsies in GBM.

We have added additional data regarding this subject to our review, which can be found at page 3, line 113.

References:

  1. Verhaak, R.G.; Hoadley, K.A.; Purdom, E.; Wang, V.; Qi, Y.; Wilkerson, M.D.; Miller, C.R.; Ding, L.; Golub, T.; Mesirov, J.P.; et al. Integrated genomic analysis identifies clinically relevant subtypes of glioblastoma characterized by abnormalities in PDGFRA, IDH1, EGFR, and NF1. Cancer Cell 2010, 17, 98-110, doi:10.1016/j.ccr.2009.12.020.
  2. Neftel, C.; Laffy, J.; Filbin, M.G.; Hara, T.; Shore, M.E.; Rahme, G.J.; Richman, A.R.; Silverbush, D.; Shaw, M.L.; Hebert, C.M.; et al. An Integrative Model of Cellular States, Plasticity, and Genetics for Glioblastoma. Cell 2019, 178, 835-849 e821, doi:10.1016/j.cell.2019.06.024.
  3. Patel, A.P.; Tirosh, I.; Trombetta, J.J.; Shalek, A.K.; Gillespie, S.M.; Wakimoto, H.; Cahill, D.P.; Nahed, B.V.; Curry, W.T.; Martuza, R.L.; et al. Single-cell RNA-seq highlights intratumoral heterogeneity in primary glioblastoma. Science 2014, 344, 1396-1401, doi:10.1126/science.1254257.
  4. Day, E.K.; Sosale, N.G.; Xiao, A.; Zhong, Q.; Purow, B.; Lazzara, M.J. Glioblastoma Cell Resistance to EGFR and MET Inhibition Can Be Overcome via Blockade of FGFR-SPRY2 Bypass Signaling. Cell Rep 2020, 30, 3383-3396 e3387, doi:10.1016/j.celrep.2020.02.014.
  5. Jun, H.J.; Acquaviva, J.; Chi, D.; Lessard, J.; Zhu, H.; Woolfenden, S.; Bronson, R.T.; Pfannl, R.; White, F.; Housman, D.E.; et al. Acquired MET expression confers resistance to EGFR inhibition in a mouse model of glioblastoma multiforme. Oncogene 2012, 31, 3039-3050, doi:10.1038/onc.2011.474.
  6. Turke, A.B.; Song, Y.; Costa, C.; Cook, R.; Arteaga, C.L.; Asara, J.M.; Engelman, J.A. MEK inhibition leads to PI3K/AKT activation by relieving a negative feedback on ERBB receptors. Cancer Res 2012, 72, 3228-3237, doi:10.1158/0008-5472.CAN-11-3747.
  7. Greene, C.; Campbell, M. Tight junction modulation of the blood brain barrier: CNS delivery of small molecules. Tissue Barriers 2016, 4, e1138017, doi:10.1080/21688370.2015.1138017.
  8. Pardridge, W.M. The blood-brain barrier: bottleneck in brain drug development. NeuroRx 2005, 2, 3-14, doi:10.1602/neurorx.2.1.3.
  9. Draaisma, K.; Chatzipli, A.; Taphoorn, M.; Kerkhof, M.; Weyerbrock, A.; Sanson, M.; Hoeben, A.; Lukacova, S.; Lombardi, G.; Leenstra, S.; et al. Molecular Evolution of IDH Wild-Type Glioblastomas Treated With Standard of Care Affects Survival and Design of Precision Medicine Trials: A Report From the EORTC 1542 Study. J Clin Oncol 2020, 38, 81-99, doi:10.1200/JCO.19.00367.
  10. Muller Bark, J.; Kulasinghe, A.; Chua, B.; Day, B.W.; Punyadeera, C. Circulating biomarkers in patients with glioblastoma. Br J Cancer 2020, 122, 295-305, doi:10.1038/s41416-019-0603-6.

Round 2

Reviewer 3 Report

Comments and Suggestions for Authors

Manuscript can be accepted in current form

Comments on the Quality of English Language

Minor english editing is required